# COMPARING PINNS ACROSS FRAMEWORKS: JAX, TENSORFLOW, AND PYTORCH

**Reza Akbarian Bafghi**
Department of Computer Science
University of Colorado, Boulder
`reza.akbarianbafghi@colorado.edu`

**Maziar Raissi**
Department of Mathematics
University of California, Riverside
`maziar.raissi@ucr.edu`

## ABSTRACT

Physics-Informed Neural Networks (PINNs) have become a pivotal technology for adhering to physical laws and solving nonlinear partial differential equations (PDEs). Enhancing the performance of PINN implementations can significantly quicken the pace of simulations and foster the creation of innovative methodologies. This paper presents 'PINNs-JAX', an innovative implementation that utilizes the JAX framework to leverage the distinct capabilities of XLA compilers. This approach aims to improve computational efficiency and flexibility within PINN applications. We conduct a comprehensive comparison of PINNs-JAX against traditional PINN implementations in widely-used frameworks such as TensorFlow V1, TensorFlow V2, and PyTorch, evaluating performance across a variety of six different examples. These include continuous, discrete, forward, and inverse problems. Our findings indicate that PINNs implemented with JAX outperform in simpler examples, yet TensorFlow V2 presents potential benefits for tackling large-scale challenges, as exemplified by the 3D-Navier Stokes case. To support collaborative development and further research, we have made the source code available to the public at: `https://github.com/rezaakb/pinns-jax`.

## 1 INTRODUCTION

Physics-informed neural Networks (PINNs) are emerging as a powerful approach for supervised learning that ensures solutions adhere to the laws of physics, especially nonlinear partial differential equations (PDEs) (Raissi et al., 2019), and have been applied across various domains (Haghighat et al., 2021; Rasht-Behesht et al., 2021; Mohammadian et al., 2022).

This paper introduces PINNs-JAX, a novel package based on the JAX framework, known for its just-in-time compilation of Python functions into XLA-optimized kernels (Bradbury et al., 2018). Our study conducts a thorough comparison of JAX with other frameworks to identify the most suitable platform for implementing PINNs. We compare the performance and capabilities of PINNs-JAX with those implemented in TensorFlow V1, PINNs-TF2 within TensorFlow V2 (Bafghi & Raissi, 2023a), and PINNs-Torch in PyTorch (Bafghi & Raissi, 2023b). This evaluation is aimed at providing researchers with a comprehensive comparison to guide their choice of the optimal framework for their specific applications.

Building on prior JAX implementations of Physics-informed Neural Networks (PINNs) (Stanziola et al., 2021; Wang et al., 2023; Sung et al., 2022) and drawing lessons from previous packages (McClenny et al., 2021; Hennigh et al., 2020; Lu et al., 2021), PINNs-JAX leverages XLA and JIT compilers for high-performance numerical computing and the construction of static computational graphs, aiming to enhance PINNs' computational efficiency. Our package is demonstrated through six examples, showcasing significant speed improvements compared to TensorFlow V1 implementations and performance across various frameworks. Furthermore, applying our approach to a large-scale real-world problem sheds light on how batch sizes and the number of trainable parameters influence the performance of the package, suggesting that TensorFlow V2 may be the preferable option for large-scale challenges.

## 2  PINNs-JAX Package

In this section, we provide a succinct overview of the problem setup utilized in PINNs and outline the functionality of our package by detailing the workflow of our package and the use of the XLA compiler in JAX.

### 2.1  Problem Setup

We adopt the problem framework from (Raissi et al., 2019), focusing on parametric and nonlinear PDEs of the form:

$$u_t + \mathcal{N}[u; \lambda], \quad x \in \Omega, \quad t \in [0, T]$$

where $u(t, x)$ is the sought solution within domain $\Omega \subset \mathbb{R}^D$, and $\mathcal{N}[.; \lambda]$ is a nonlinear operator dictated by parameter $\lambda$. The study explores two main challenges: forward problems, which involve deducing the system's hidden state $u(t, x)$ for given $\lambda$ (Wang et al., 2022; Raissi et al., 2016), and inverse problems, focused on identifying the parameters $\lambda$ that best match the observed data (Raissi et al., 2017; Raissi & Karniadakis, 2017; Rudy et al., 2016). The research develops algorithms for both continuous and discrete time models, using new approximators for the former and Runge-Kutta methods for the latter, detailed in (Raissi et al., 2019).

### 2.2  Implementation

**PINNs-JAX Workflow.**  Our package enhances the PINNs framework from (Bafghi & Raissi, 2023b) for solving PDE-related forward and inverse problems using Hydra (Yadan, 2019), adding support for new boundary conditions, such as inlet, outlet, upper, and lower walls in 2D space and features like prediction saving. In summary, it processes configuration files to set up domains, sampling, and neural network configurations. The process involves reading user-defined PDEs and configurations, compiling conditions, and capturing a computational graph with the XLA compiler for efficient training. For optimization, we employ Optax, a flexible gradient processing and optimization library for JAX, developed by (DeepMind et al., 2020).

**XLA Compiler.**  The JAX system functions as a just-in-time (JIT) compiler, generating code for subroutines that are pure and statically composed—meaning a function is pure if it has no side effects and is statically composed if it can be represented as a static data dependency graph based on a set of primitive functions. It achieves this through high-level tracing in conjunction with the XLA compiler infrastructure (Demeure et al., 2023). JAX builds upon the tracing library used by Autograd (Maclaurin, 2016), which is designed for self-closure and thus recognizes its operations as primitives. Additionally, JAX incorporates Numpy's (van der Walt et al., 2011) numerical functions as part of its primitives, enabling it to generate code for Python functions that use Numpy syntax. This includes supporting arbitrary-order forward and reverse-mode automatic differentiation (Frostig et al., 2018).

## 3  Experiments

In this section, we explore the experiments designed to assess the performance of PINNs-JAX across different scenarios, with a particular focus on how varying batch sizes impact its effectiveness. Additionally, we benchmark its performance against other popular frameworks like TensorFlow and PyTorch, aiming to identify the most efficient approach for implementing PINNs. Throughout these experiments, the Adam optimizer was exclusively utilized.

**Hardware Setup.**  All experiments were performed using a single NVIDIA Quadro RTX 8000 GPU to guarantee consistency and facilitate reproducibility.

**Speed-up Metric.**  Following (Bafghi & Raissi, 2023a;b), we calculate the median iteration time for different scenarios and compare it to that of the original TensorFlow V1 (TF1) implementations. Speed-up is measured by dividing TF1 times by the times for each scenario.

Table 1: Comparison of average speed-ups across six examples discussed in Section 3.1 using various libraries to TensorFlow V1. The table indicates that JAX outperforms compared to other libraries.

|  | PyTorch | TensorFlow V2 | JAX |
|---|---|---|---|
| Avg. speed-up w.r.t. TensorFlow V1 | 5.43 | 18.12 | **23.68** |

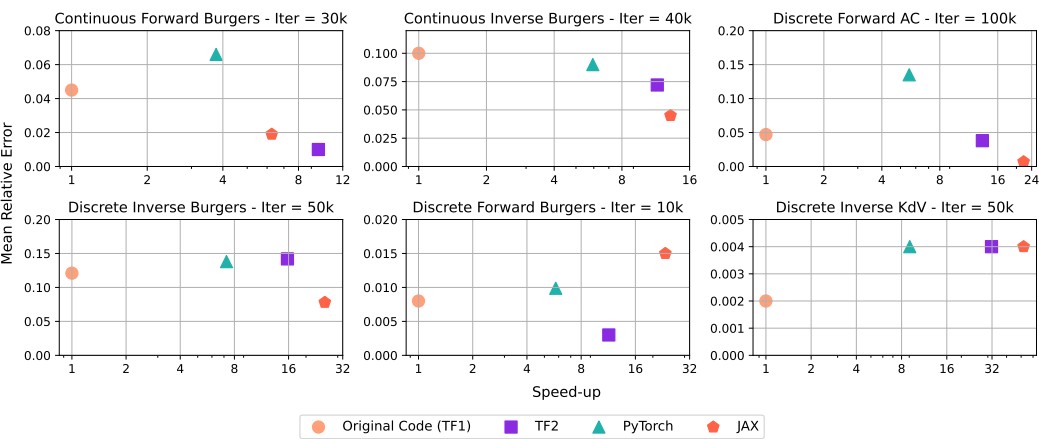

Figure 1: Each subplot represents a distinct problem, with the specific iteration count displayed at the top. The logarithmic x-axis illustrates the speed-up relative to TensorFlow V1, while the y-axis measures the mean relative error. The plot underscores that JAX enhances speed without introducing substantial error in all experiments, with the exception of Continuous Forward Burgers. Additionally, it showcases the capability of the XLA compiler, utilized by both JAX and TensorFlow, to accelerate PINNs more effectively than PyTorch.

**Frameworks.** Our experiments utilize optimal configurations for TensorFlow V2 and PyTorch as described in (Bafghi & Raissi, 2023a;b). Specifically, TensorFlow V2 models employ the XLA compilers (Sabne, 2020), whereas PyTorch models make use of CUDA Graph (Ramarao, 2022) and TorchScript (DeVito, 2022) for enhanced performance. Additionally, TensorFlow V1 does not utilize any acceleration technologies, and the original setup from the referenced work was adopted. In this paper, we exclude configurations utilizing Mixed Precision and those not employing JIT compilers due to their poor performance in PINNs (Bafghi & Raissi, 2023a;b).

## 3.1 EVALUATION OF VARIOUS FRAMEWORKS

We assess the performance of multiple frameworks across various scenarios, including the Discrete Forward Allen–Cahn (AC) Equation, and Discrete Inverse Korteweg-de Vries (KdV) Equation. Additionally, we explore Burgers' Equation in all configurations: continuous, discrete, forward, and inverse. Each example utilizes static batches. For detailed insights about the examples, refer to (Bafghi & Raissi, 2023a; Raissi et al., 2019). Our evaluations cover TensorFlow V2, PyTorch, and JAX, comparing them against a baseline in TensorFlow V1.

Figure 1 illustrates that employing JAX and TensorFlow V2 results in a significant speed enhancement over TensorFlow V1 and PyTorch when using a single GPU, without compromising accuracy. The most substantial speed-up recorded is 51.94, achieved in the KdV example. This performance boost may be attributed to JAX's functional programming nature, eliminating the need to compute the second derivative and allowing for direct calculation of the third derivative of the output. Although JAX outperformed TensorFlow V2 in most cases, the speed-up observed in the Continuous Forward Burgers example was lower with TensorFlow V2, highlighting the impact of batch size, a topic to be explored in the subsequent section. Table 1 presents the average speed-up of our implemented examples compared to TensorFlow V2 and PyTorch, showing that JAX, on average, achieves a speed-up of 23.68 in the examples mentioned.

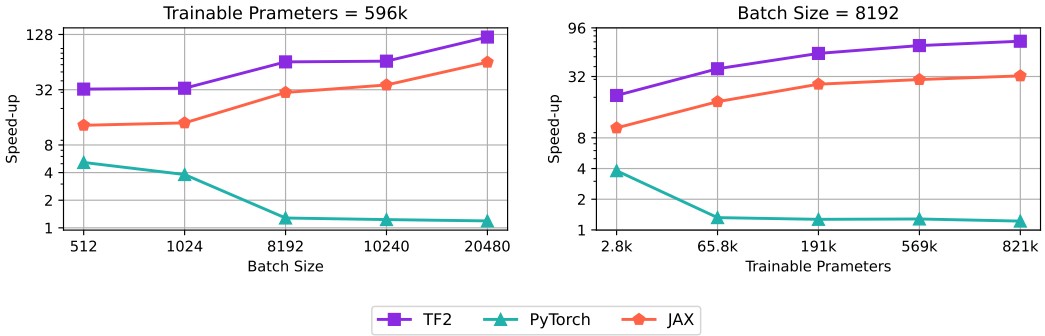

Figure 2: The left plot illustrates the speed-up achieved by various frameworks as batch sizes increase. The right plot demonstrates the performance of these libraries across varying numbers of trainable parameters, achieved by adjusting the depth of the neural network's layers. The findings indicate that TensorFlow V2, utilizing XLA compilers, surpasses the performance of other frameworks. However, JAX displays a competitive speed-up relative to TensorFlow V2.

## 3.2 Assessing the Impact of Batch Size and Number Trainable Parameters

In this subsection, we analyze the effects of batch size variations and the number of trainable parameters on model efficiency across different frameworks. Our study is centered on simulating three-dimensional physiological blood flow using a realistic intracranial aneurysm (ICA) model, governed by the 3D Navier-Stokes equation. The dataset comprises 29 million data points spanning spatial and temporal domains for five different solutions, employing dynamic batches and random sampling in each iteration. For further information on the example, we refer readers to (Raissi et al., 2018; 2020).

We evaluate speed-up metrics across various configurations, starting by modifying only the batch size while keeping all other factors constant in a model with 596k trainable parameters. The left plot in Figure 2 indicates that while JAX's performance improves with increasing batch sizes, it consistently falls short of TensorFlow V2, demonstrating TensorFlow V2's superior capability in managing large-scale problems. This enhanced performance with TensorFlow V2 and JAX with respect to PyTorch could be attributed to the XLA compiler's ability to optimize memory usage in computational graphs, for instance, through operation fusion, thereby accommodating larger batch sizes. In a subsequent experiment, with a fixed batch size of 8192, we varied the number of trainable parameters by altering the depth of the neural network's layers. Here, while performance generally decreased across PyTorch, the efficiency of XLA compilation exhibited an increase, as depicted in the right plot of Figure 2. This discussion highlights the XLA compiler's benefits, particularly for scenarios involving large batch sizes and numerous trainable parameters, and suggests that TensorFlow V2 is the more suitable library for large-scale computational tasks.

## 4 Conclusions and Limitations

In this study, we investigate the performance of Physics-Informed Neural Networks (PINNs) across various libraries, including PyTorch, TensorFlow, and JAX. We demonstrate that the use of XLA compilers in TensorFlow and JAX significantly enhances speed, highlighting the critical role of compilers in improving efficiency beyond standard TensorFlow V1 implementations. The evaluation of six examples reveals that JAX achieves a higher average speed-up of 23.68 compared to TensorFlow V2 and PyTorch. However, TensorFlow exhibits superior performance for large batch sizes and a higher number of parameters. Furthermore, by applying PINNs to a range of problems, we showcase the adaptability of the PINNs-JAX package to diverse scientific challenges. Additionally, our findings suggest that the choice of library and computational strategies can be crucial in optimizing PINNs for specific applications, reinforcing the need for a nuanced approach to their implementation. This paper aims to serve as a valuable resource for researchers and practitioners implementing PINNs across different libraries, guiding them in selecting appropriate tools and techniques for their computational needs.

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
