# OpenReview forum: "Comparing PINNs Across Frameworks: JAX, TensorFlow, and PyTorch"
_ICLR.cc/2024/Workshop/AI4DiffEqtnsInSci — AI4DiffEqtnsInSci @ ICLR 2024 Poster_

### Official Review · Reviewer_z6tM · 2024-02-26
**This study presents the computational library package leveraging the JAX framework to PINN application and conducts a comprehensive benchmarking analysis against major deep learning libraries.**

**Rating:** 4
**Confidence:** 5

**Review:**

The paper is less clear in describing how PINNs-JAX contributes to speeding up processes beyond incorporating Hydra. A clearer delineation of the novel aspects of the work should be highlighted. Additionally, it remains unclear whether the authors made improvements to the JAX compiler. But, in general, the development of a computational library optimizing PINNs addresses an important problem and has the potential to benefit the entire community. Highlighting the unique contributions of PINNs-JAX and clarifying any enhancements to the JAX compiler would enhance the quality of the work.

The study conducts a comprehensive benchmarking of standard DNN libraries for various PINN applications, thereby raising awareness within the community regarding the trade-offs between speed and model sizes. Furthermore, it provides insights into the further development of DNN libraries. However, it remains unclear how PINNs-JAX differs from the performance achieved by the original JAX library. Addressing this aspect would improve the significance of PINNs-JAX proposed in this study in the context of existing libraries and contribute to a clearer understanding of its potential impact.

---

### Official Review · Reviewer_J8p8 · 2024-02-26
**Review of Comparing PINNs Across Frameworks: JAX, TensorFlow, and PyTorch**

**Rating:** 8
**Confidence:** 3

**Review:**

The paper implements Physics-Informed Neural Networks using a JAX framework and show that this implementation is more efficient than Pytorch and Tensorflow for simple problems. The authors support this finding with several experiments. This paper is well written and clear, and there are no notable weaknesses.

---

### Meta-Review · Program_Chairs · 2024-03-03

**Recommendation:** Accept (Poster)

**Metareview:**

I agree with reviewer Reviewer J8p8 and also vote for acceptance. I encourage the authors to address the comments from Reviewer z6tM in the camera-ready version.

---

### Decision · Program_Chairs · 2024-03-03

Accept (Poster)